

# Co-circulation of all the four Dengue virus serotypes during 2018–2019: first report from Eastern Uttar Pradesh, India

Sthita Pragnya Behera[1,*], Pooja Bhardwaj[1,*], Hirawati Deval[1], Neha Srivastava[1], Rajeev Singh[1], Brij Ranjan Misra[1], Awdhesh Agrawal[2], Asif Kavathekar[1] and Rajni Kant[1]

[1] ICMR-Regional Medical Research Centre, Gorakhpur, Uttar Pradesh, India
[2] Division of Pathology, Gorakhnath Hospital, Gorakhpur, Uttar Pradesh, India
* These authors contributed equally to this work.

## ABSTRACT

Dengue fever is an endemic disease in India, transmitted by an infected mosquito bite. In India, the number of concurrent infections and the circulation of multiple dengue virus (DENV) serotypes has increased in recent decades. Molecular surveillance among the DENV serotype is important to keep track of the circulating serotypes, evolutionary changes, and key mutations that can alter the diagnostics. The current study included patients admitted for dengue in the Eastern Uttar Pradesh (E-UP) region during 2018–2019. The genetic characterization of the circulating DENV was accomplished through partial CprM (511 bp) gene amplification *via* reverse transcriptase polymerase chain reaction and sequencing. Phylogenetic analysis revealed the circulation of all four DENV1-4 serotypes. DENV-2 was the most abundant serotype (44%, 27/61), followed by DENV-3 (32%, 20/61). DENV-1 had a 16% (10/61) predominance, while DENV-4 (6%, 4/61) was found to be the least abundant serotype. DENV-2 genotypes were distributed among lineages I (7.4%), II (85%) and III (7.4%) of genotype IV, DENV-3 to lineage III of genotype III, DENV-1 to genotype III; DENV-2 to lineage B (75%) and C (25%) of genotype I. This primary report on the co-circulation of DENV1-4 serotypes from the E-UP region highlights the requirement of continuous molecular surveillance for monitoring circulating DENV serotypes.

# INTRODUCTION

Dengue (DEN) is a viral disease caused by any of the four serotypes of the Dengue virus (DENV) and transmitted through an infected Aedes mosquito bite (*A. aegypti or A. albopictus*). After malaria, it is the second most frequent mosquito-borne disease that affects humans (*World Health Organization, 2009*). During the past decades, the trends of DENV transmission has been shifted from urban to peri-urban and rural areas as well (*Chakravarti, Arora & Luxemburger, 2012*; *Bhatt et al., 2013*; *Ganeshkumar et al., 2018*; *Murhekar et al., 2019*). Based on the amino acid variations ~25 to 40%, DENV has been categorized into four antigenically related serotypes DENV1-4. Further, 3% of

Corresponding author
Hirawati Deval,
dr.hirawati@gmail.com

aminoacid-level variations classify each DENV serotype into genotypes (*Dowd, DeMaso & Pierson, 2015*). The fifth serotype DENV-5 has also been reported in the last decade following the sylvatic life cycle (*Mustafa et al., 2015*). However, vaccines for DENV have been available now and various are under the process of development. However, to date, no effective vaccine has been developed that can target all four DENV serotype neutralization (*Kukreti et al., 2010*; *Diamond & Pierson, 2015*).

Globally, India is an endemic country for DENV and the primary cause of hospitalization. The majority of Indian states have been categorised as having a high or constant risk of dengue transmission (*Murhekar et al., 2019*). Furthermore, molecular and sero-surveillance demonstrated the circulation of all DENV1-4 serotypes, co-circulation of multiple serotypes, and concurrent infection from DENV serotypes from different geographical regions of India (*Mishra et al., 2017*; *Ganeshkumar et al., 2018*; *Alagarasu et al., 2021*).

In India, as of 2021 Uttar Pradesh (UP) is marked as the most populated state and due to overcrowding and poor hygiene, DENV transmission in UP has raised a major health concern. Further, DENV is also reported to be an etiological agent for acute encephalitis syndrome (AES) (*Vasanthapuram et al., 2019*). The study carried out with hospitalized patients from 2015 to 2016 in eastern UP (E-UP) and adjoining areas of Bihar demonstrated the circulation of DENV-2 serotypes lineages I, II and III (*Deval et al., 2021*). The continuous monitoring of DENV transmission is required for an efficient DENV control program. The capsid pre membrane (CprM) junction of the DENV viral genome has been suggested to be time and cost-efficient for genotyping with a single primer set for both amplification and sequencing (*Mishra et al., 2015*; *Dieng et al., 2021*). Furthermore, various literature had cited the use of the CprM region for phylogenetic analysis and genetic characterization of DENV serotypes. Hence, in the present retrospective study, the CprM gene junction was targeted to investigate the genotype of circulating DENV serotype in E-UP during the October and November months of the years 2018 and 2019.

## MATERIALS AND METHODS

### Sample collection

The study was conducted on patients admitted at Gorakhnath Tertiary care hospital, Gorakhpur (having a platelets transfusion facility) during the post-monsoon period of 2018 and 2019. Two ml of blood was collected from each DEN suspected patient after obtaining written consent. The initial diagnosis was performed in the hospital setting using a rapid diagnostic kit (J. Mitra and Co Pvt Ltd, New Delhi, India) that identifies DEN NS1 antigen and anti-dengue IgM/IgG antibodies. Further, aliquots of sera (0.5 ml) collected from DEN NS1-positive cases were transported to the ICMR-Regional Medical Research Centre, Gorakhpur (RMRC) laboratory for further confirmation by reverse-transcription polymerase chain reaction (RT-PCR). According to the manufacturer's instructions, the sera were analysed using a DEN NS1 antigen ELISA (J. Mitra and Co Pvt. Ltd, New Delhi, India) designed for the *in-vitro* qualitative detection of DEN (serotypes 1–4) NS1 antigen in human serum or plasma.

These investigations were approved and carried out following the guidelines established by the IHEC, RMRC, Gorakhpur (RMRCGKP/EC/2022/3.17).

## Viral RNA extraction and RT-PCR

QIAamp Viral RNA Mini Kit (Qiagen, Hilden, Germany) was used to extract viral RNA from 140 μL sera according to the manufacturer's instructions. RNA was eluted in 50 μL of the AVE buffer provided and stored at −80 °C until needed. Isolated RNA was then subjected to RT-PCR using *Promega Access RT-PCR kit*, (Promega, Madison, WI, USA) containing consensus forward primer DENF1 5′-TCAATATGCTGAAACGCGCGAGA AACCG-3′ and reverse primer DENC1 5′-TTGCACCAACAGTCAATGTCTTCAGGT TC-3′ targeting the 511 nucleotide fragment spanning capsid pre membrane (CprM) junction of all the four serotypes of Dengue virus (*Lanciotti et al., 1992*). The thermal cycling conditions of the RT-PCR reaction include a RT step at 50 °C for 30 min followed by an initial denaturation at 94 °C for 3 min, then subjected to 35 cycles of denaturation at 94 °C for 30 s, annealing at 57 °C for 45 s, and extension at 72 °C for 2 min, with the final extension of 72 °C for 10 min. A negative control was included at every reaction setup where the template has been replaced with water to serve as no template control.

## Sequencing and phylogenetic analysis

The amplicon generated by RT-PCR was visualized by SyBr safe nucleic stain (Thermo Fisher Scientific, Waltham, MA, USA). The samples harbouring 511 bp target amplicon were then gel extracted using Quaquick Gel extraction kit (Qiagen, Hilden, Germany). DNA sequencing from both ends of the CprM gene was conducted on an ABI 3130 DNA sequencer (ABI) using the BigDye Terminator Cycle Sequencing Ready Reaction Kit (ABI) as previously described (*Deval et al., 2021*).

The alignment of partial CprM sequences from this study and references sequences retrieved from GenBank (GB) was performed using the ClustalW. The alignment of partial CprM sequences generated from this study and references sequences retrieved from GenBank (GB) was performed using the ClustalW platform. The aligned sequences were trimmed manually to obtain the consensus sequence. The phylogenetic tree was constructed through Molecular Evolutionary Genetics Analysis (MEGA) software version X. The evolutionary history was inferred by using the Maximum Likelihood method based on the Tamura-Nei model. The reliability of the tree was estimated using 1,000 bootstrap replications under the Nearest-Neighbour interchange procedure with input distance determined by the Maximum-Likelihood method (*Kumar et al., 2018*).

## RESULTS

During 2018 and 2019 DENV seasons, a total of 164 serum samples that tested positive from the NS1 fast antigen kit were received from Gorakhnath Tertiary care hospital, Gorakhpur. These 164 serum samples were subjected to RT-PCR targetting the CprM region of DENV. Of these 164, 61 (37.1%) serum samples tested positive for DENV genome. Further of these 61 samples, 24.5% (15/61) were found to be positive for IgM and 18% (11/61) for IgG ELISA. Among these RT-PCR positive tested samples, the majority

**Table 1 Demographics parameters of the patients tested positive for DENV genome.**

| S. no. | Characteristics | Number ($n = 61$) | Percentage (%) |
|---|---|---|---|
| 1 | Age (Years) | | |
| | 0–6 | 1 | 1.6 |
| | 7–20 | 15 | 24.6 |
| | >18 | 45 | 73.7 |
| 2 | Gender | | |
| | Male | 48 | 78.68 |
| | Female | 13 | 21.31 |
| 3 | ELISA | | |
| | IgG ELISA | 11 | 24.5 |
| | IgM ELISA | 15 | 18.0 |

was comprised of male (78.68 %), and male ($n = 48$) to female ($n = 13$) ratio was found to be 1:0.2. The age of the patients ranged from 4 to 62 years comprising 1.6% of 0–7, 24.6% 8–17 and 73.7% of ≥18 years of age (Table 1).

The sequences obtained from these 61 cases were further analyzed for serotyping analysis. The 61 cases were distributed among DENV1 to DENV-4 serotypes (Fig. 1). Among which 27 cases were found to be of DENV-2 (44%), 20 of DENV-3 (32%), 10 of DENV-1 (16%) and 4 of DENV-4 (06%). The results demonstrated clearly that all four serotypes were co-circulated during the 2018-2019 DENV season in E-UP with the prevalence of the DENV-2 serotype. Followed which the DENV-3 serotype was found to be pre-dominant while the patients were also found to be infected by other DENV serotypes. The sequences ($n = 61$) retrieved from this study were submitted to GenBank (GB) under the accession (acc.) number (no.) MZ490477–MZ490537.

To identify the genotypes among each DENV serotype, CprM gene sequences obtained during this investigation, as well as sequences from different geographical sites around the world, were retrieved from the NCBI database and used for phylogenetic analysis. Genotyping analysis of DENV serotypes revealed the distribution of DENV-1 serotypes among five genotypes including sylvatic and genotype I–IV. In this study DENV-1 isolates ($n = 10$) were classified as genotype III and clustered into two groups separately in the dendrogram (Fig. 2). Cluster 1 containing seven sequences was associated with DENV-1 sequences isolated from different states (West Bengal, Maharashtra, Karnataka) in the country whereas cluster 2 with 3 isolates showed similarity with the strains isolated from the adjoining bordered state (Madhya Pradesh; MH051271) and same province *i.e.*, Lucknow.

The DENV-2 serotypes were distributed among six genotypes including sylvatic and genotype I–V. The 27 strains from this study of DENV-2 were classified as cosmopolitan (Genotype IV) and were distributed among Lineage II (85%, $n = 23$) and 7.5% ($n = 2$) among lineage I and III individually. A total of 17 isolates from the lineage II shared similarity with isolates from India, Singapore and China, whereas the five strains (MZ490488, MZ490494–MZ490495 and MZ490509–MZ490510) showed no significant

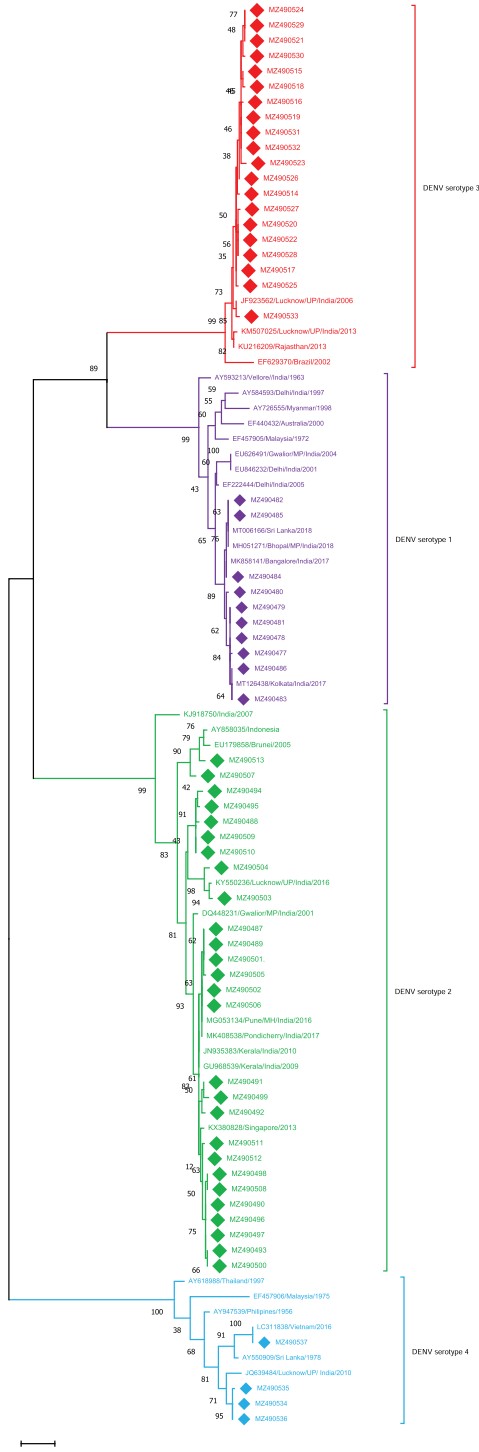

**Figure 1 Serotyping of DENV isolates using CprM gene sequences ($n$ = 61) from E-UP.** The tree is based on partial CprM gene sequences. The numerical value on nodes represents percentage of 1,000 bootstrap value. In the tree DENV-1 subtree is shown with purple nodes, DENV-2 with green nodes, DENV-3 with red nodes, and DENV-4 with sky blue nodes. The sequences from this study are written in GenBank (GB) accession number and highlighted with filled diamonds. Other strains are highlighted with GB number, country and year of specimen collection or virus isolation.

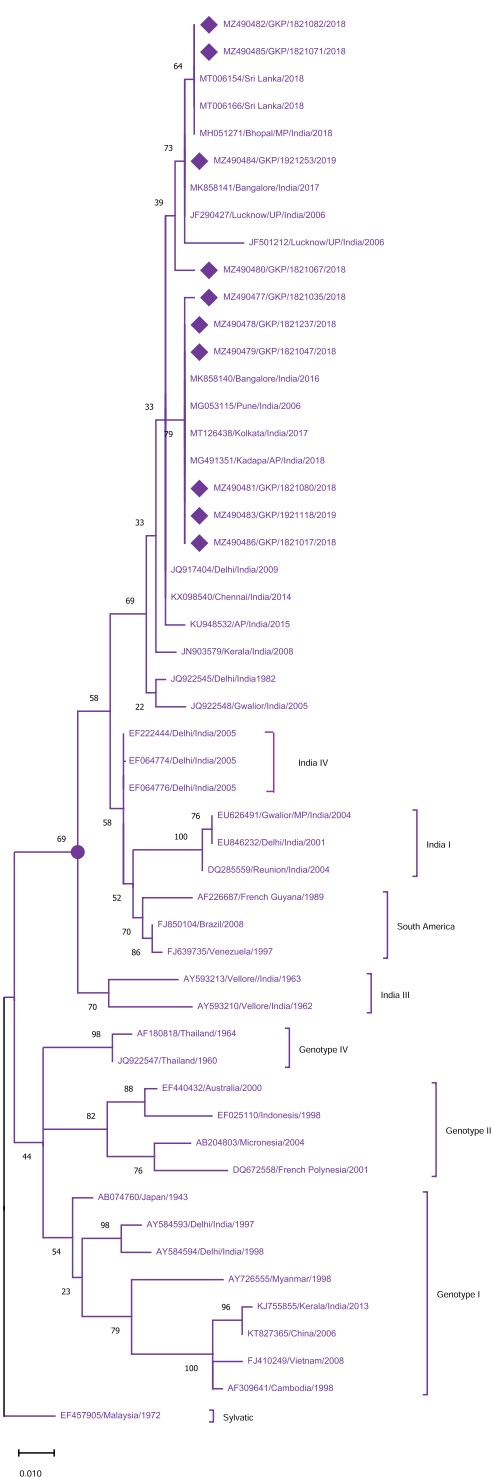

**Figure 2 Genotyping of DENV-1 serotype isolates using CprM gene sequences (*n* = 10) from E-UP.**
The strains from this study are shown with filled diamond followed by Genbank accession, isolate ID and
year of isolation. The reference strains are indicated by Genbank accession number followed by country
and year of isolation. Numerical value at nodes represents the bootstrap value of 1,000 replicates.
The subtree with circle at node presents Genotype III cluster of DENV-1 serotype.

similarity to any of the reference strains and were clustered separately in lineage II (Fig. 3). The E-UP isolates shared 99.9 ± 0.09% nucleotide identity among and across the DENV-2 genotypes. Six isolates from 2019 season showed similarity with Indian strains, 2018 isolates ($n = 3$) showed similatity with Delhi strains from lineage II. Remarkably the E-UP isolates in lineage I and III ($n = 2$) belongs to 2018 and 2019 season respectively, where 2019 isolates shared similarity with Lucknow strains and 2018 isolate shared similarity with Brunei (2005) isolate.

As depicted in Fig. 4 the DENV-3 serotypes were distributed among four genotypes namely genotype I, II, III and V. The E-UP isolates ($n = 20$) of DENV-3 shared ≈99.9% nucleotide identity and were classified as genotype III and distributed into two clusters under Lineage III (Fig. 4). All E-UP strains from cluster one; of DENV-3 showed China (2013) (KF954947) and Singapore strain (2012) except one. A single isolate from 2019 season (MZ490533) shared the closest similarity with the Lucknow, UP strain (JF923562).

Genotyping analysis of DENV-4 serotypes demonstrated the distribution of isolates in five genotype including genotype I–V. The DENV-4 serotypes ($n = 4$) detected from this region were classified as genotype I and all strains shared similarity with the Indian strains (Lineage C) whereas one strain MZ490533 was placed in a different cluster (Lineage B) which was not reported earlier from India. MZ490533 was identified as 2018 isolate and shared closest similarity with Vietnam (2016). The three E-UP isolates from Lineage C shared 99.99% nucleotide identity but showed 0.08% nucleotide difference with MZ490533 (Fig. 5). Overall the DENV strains identified during 2018–2019 season were closely related to neighbouring states or countries which establishes that DENV has been introduced to E-UP region from adjoining state or within the state or through the neighbouring countries.

## DISCUSSION

UP is the 4th largest and one of the most densely populated states of India. The E-UP region is a geographical sub-region of Uttar Pradesh state that touches the international boundary with Nepal in the North. The E-UP region is endemic to acute encephalitis syndrome (AES) (*Kakkar et al., 2014*). DEN serves as an important and common etiological agent for AES cases other than Japanese encephalitis and Scrub typhus, responsible for contributing 5.2% alone of the total AES cases in Uttar Pradesh (*Vasanthapuram et al., 2019*). Due to the severity caused by the disease, NVBDCP of India; the nodal agency for AES surveillance in India, has recently added serological testing for DENV to the national AES testing algorithm. The heightened incidences of DENV outbreak and the additional etiological agent of AES in the E-UP region prompted us to do the continuous surveillance of DENV from a public health point of view.

Most of the DENV epidemiological studies were conducted either from western UP (W-UP) or from central UP, while limited data are available from the E-UP region. In 1968, the first epidemic of DENV was reported in Kanpur located in the W-UP region, followed by several epidemics in 1969, 1996, 2003, 2004, 2006, and 2008 onwards (*Gupta et al., 2012*; *Pandey et al., 2012*; *Prakash et al., 2015*). The Kanpur outbreak was due to DENV-4, while both DENV-2 and DENV-4 were responsible for the outbreak in 1969.

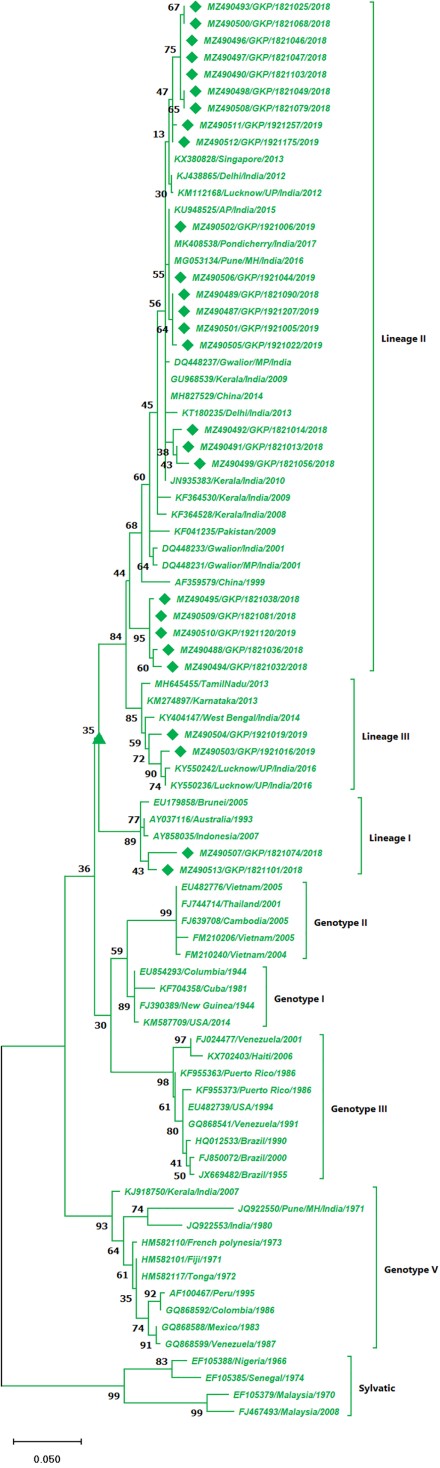

**Figure 3 Genotyping of DENV-2 serotype isolates using CprM gene sequences ($n$ = 27) from E-UP.**
The strains from this study are shown with filled diamond followed by Genbank accession, isolate ID and
year of isolation. The reference strains are indicated by Genbank accession number followed by country
and year of isolation. Numerical value at nodes represents the bootstrap value of 1,000 replicates.
The subtree with upright triangle at node presents Genotype II cluster of DENV-2 serotype.

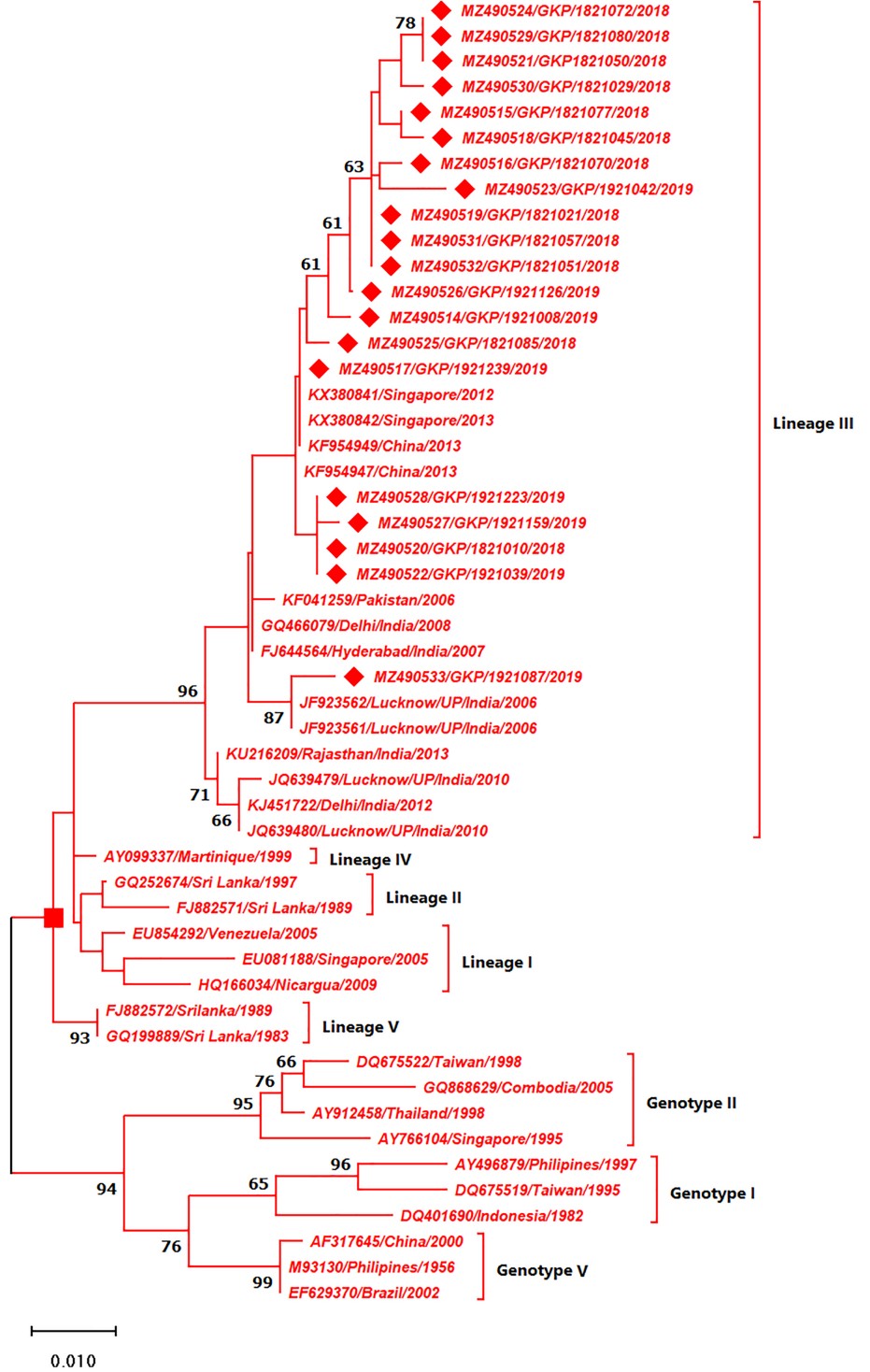

**Figure 4 Genotyping of DENV-3 serotype isolates using CprM gene sequences (*n* = 20) from E-UP.** The strains from this study are shown with filled diamond followed by Genbank accession, isolate ID and year of isolation. The reference strains are indicated by Genbank accession number followed by country and year of isolation. Numerical value at nodes represents the bootstrap value of 1,000 replicates. The subtree with square at node presents Genotype III cluster of DENV-3 serotype.

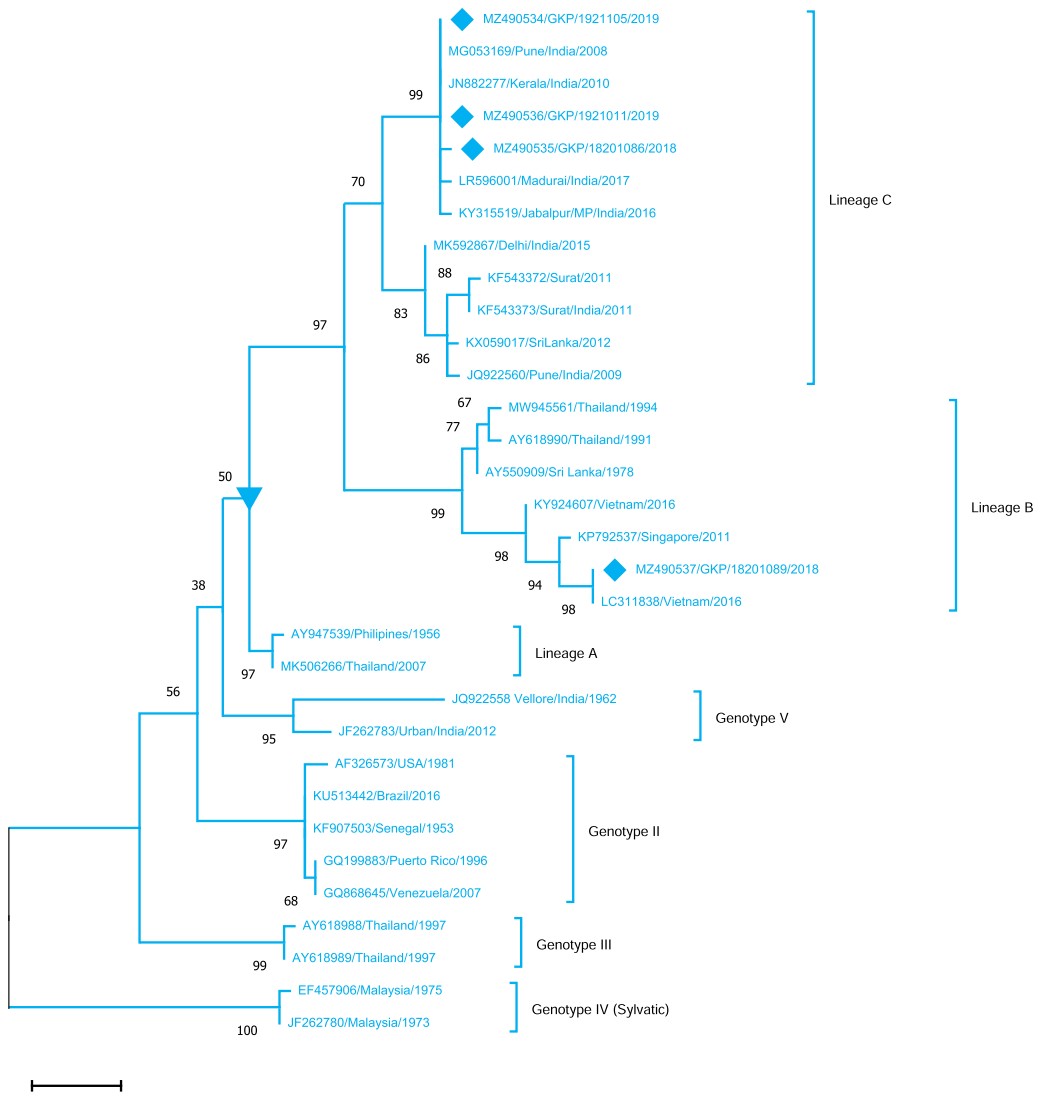

**Figure 5 Genotyping of DENV-4 serotype isolates using CprM gene sequences (*n* = 4) from E-UP.** The strains from this study is shown with filled diamond followed by Genbank accession, isolate ID and year of isolation. The reference strains are indicated by Genbank accession number followed by country and year of isolation. Numerical value at nodes represent the bootstrap value of 1,000 replicates. The subtree with inverted triangle at node presents Genotype I cluster of DENV-4 serotype.

Further, *Mishra (2021)* found co-circulation of three serotypes DENV (DENV-1, DENV-2 and DENV-3) from 2009 to 2012 from the Central UP region. According to the National Vector Borne Disease Control Programme (NVBDCP), an erractic number of dengue cases (23,128 cases) as of November 2021 have been reported mostly the pediatric population. Authors assumes that co-circulation, concurrent infection and shift in genotype lineage of predominant serotype or replacement of existing serotype by other resulted 2021 DEN outbreak in W-UP, But this assumption needed to be tested further (*Mishra, 2021*).

The study carried out during past DENV outbreak (2015–2016) in the E-UP region detected the prevalence of only DENV-2 serotype genotyping based on the Envelope (E) gene (*Deval et al., 2021*). However, the structural genes including E, E-NS1 junction, and CprM are commonly utilised for DENV serotyping and genotyping (*Dieng et al., 2021*). Phylogenetic trees based on CprM and E gene areas have the same topology across genotypes/sublineages (*Yergolkar et al., 2017*). Hence, the current investigation has been carried out using CprM, as the larger dataset is available for CprM gene sequences of Indian DENV isolates. In the current DENV epidemiological study, serotyping and genotyping based on the CprM region were done, and we found the co-circulation of all four DENV serotypes with the prevalence of 44% DENV-2 ($n = 27$), 32% DENV-3 ($n = 20$), 16% DENV-1 ($n = 10$), and 6% DENV-4 ($n = 4$).

The cosmopolitan genotype (genotype IV) of DENV-2 has outspread distribution in the tropics and subtropics. Moreover, genotype IV of DENV-2 was associated with outbreaks in different parts of Northern India such as New Delhi, Lucknow, Gwalior and E-UP (*Parida et al., 2002*; *Pandey et al., 2012*; *Islam et al., 2016*). Our earlier study from this region found the DENV outbreak that occurred between 2015 and 2016 was solely due to the cosmopolitan genotype of DENV-2. Also during the dengue outbreak in 2015–2016, we found a shifting of lineage from (lineage I and lineage III) in 2015 to lineage II in 2016. In continuation with the earlier study, after 2 years of interval, we found that lineage II of cosmopolitan genotype is still the predominant DENV-2 serotype prevailing in this region indicating the above genotype of DENV has not undergone much evolution and resting passively. Further from the phylogenetic analysis, we found the lineage II DENV-2 sequences of genotype II were found genetically identical to earlier reported outbreak strains of DENV-2 from New Delhi, Gwalior, Kerala, and Pune. Recent, study found extensive transportation, micro-evolution and circulation of more than one lineage of DENV-2 genotype IV, are the factors that contributed to the dominant emergence of DENV-2 in different states of India (*Alagarasu et al., 2021*).

Interestingly DENV-3 emerged as a leading serotype after DENV-2 in this region. DENV-3 cases were distributed under lineage III and showed close homology with DENV-3 lineages III strains from Lucknow isolated during 2006 (JF923562), 2010 (JQ639479) and from New Delhi during 2008 (GQ466079) and 2012 (KJ451722). Amongst other genotypes of DENV-3, genotype III (lineage III) of DENV-3 viruses is predominant and is the major cause of dengue infection in India (*Sharma et al., 2011*). Further, this genotype of DENV-3 was responsible for the outbreak during 2003–2004 in Northern India, particularly in New Delhi and Gwalior, which has replaced the earlier circulating cosmopolitan genotype of DENV-2. Similar incidence could happen in this region too, as the cosmopolitan genotype of DENV-2 and subtype III of DENV-3 are prevailing and DENV-3 is emerging, therefore public health officials should take preventative measures to mitigate outbreak situations or vaccine trials in near future.

As evidenced from the previous studies genotypes I (GI) & III (GIII) of DENV-1 are the predominant genotypes circulating in India since 1956 (*Kukreti et al., 2008*). Furthermore, DENV-1 GIII circulation was previously reported from the central UP between 2009 and 2012 (*Mishra et al., 2015*). In line with this, our findings showed that all DENV-1

sequences from this study clustered within GIII along with strains reported from different parts of India including Pune, Bangalore, Kolkata, Bhopal and Central UP (Lucknow). Our findings, together with other findings, indicate that GIII of DENV-1 is the dominant genotype of UP. During 2012, GIII of DENV-1 was responsible for an outbreak in neighbouring states Chhattisgarh and Madhya Pradesh (*Barde et al., 2015*). In addition, similar to the genotypic/lineage replacement of DENV-2, DENV-1 GIII strains were replaced by DENV-1 GI strains, in Southern India. This lineage shift was responsible for the DENV-1 outbreaks in 2012–2015 (*Fazil et al., 2019*). Therefore, continuous monitoring is required in dengue-endemic areas such as E-UP to track down the introduction of any new genotype/lineage to limit future dengue outbreaks.

DENV-4, the least identified serotype, has been genetically classified as genotype I (GI) in our study as well as in other Indian studies from South India, Delhi, Maharashtra, Telangana, and Andhra Pradesh (*Dash et al., 2011*; *Cecilia et al., 2017*; *Racherla et al., 2018*; *Shrivastava et al., 2018*; *Fazil et al., 2019*). This is possibly the first report of DENV-4 detection from Eastern U.P., India. Furthermore, DENV-4 GI serotypes discovered in this study were distributed between lineage B ($n = 1$) and lineage C ($n = 3$). DENV-4 lineage B has only been reported from other countries. The majority of the strain is of lineage C, which is consistent with earlier research from India. While one strain (MZ490533) corresponds to lineage B, which has only been reported from other countries, we believe that this sequence originated from different nations, showing that trans-border infection may have happened as a result of travel. Furthermore, the seeding and dispersal of DENV1, DENV-2 and DENV-3 lineages across Asia have been demonstrated to be strongly linked with Air traffic (*Tian et al., 2017*). The detection of all serotypes in our investigation confirms active DENV co-circulation in this location. While DENV-2 activity has already been detected in the same area.

Additionally, we have marked that the serotypes isolate from the 2019 season demonstrated close relationship with the Indian isolates, whereas at least one isolate from each serotype obtained during 2018 season demonstrated similarity with strains from different countries. This demonstrated the serotype other than DENV-2 has been introduced to E-UP during 2018 and 2019 through foreign and across the country respectively. Further, when many dengue serotypes circulate in a population, the likelihood of co-infection escalates. Multi-serotype circulation zones are crucial for severe dengue infection (*Vaddadi et al., 2017*). Furthermore, co-infection with multiple serotypes may result in the emergence of recombinant virus strains with diverse characteristics (*Shrivastava et al., 2018*). However, no indication of co-infection was observed among the patients in our study.

## CONCLUSION

Despite the fact that Dengue has been present in the UP for decades, relatively few studies have been undertaken to investigate the genetic composition of the virus. As compared to past study, shift in DENV serotype prevalence from DENV-2 to multiple serotypes has been observed. The present study revealed the circulation of all four serotypes of the DENV in E-UP. In addition to GIII of DENV-1 and DENV-3, the cosmopolitan genotype

of DENV-2 remains to be the predominant circulating genotype. Additionally, DENV-4 lineages B and C are detected from this region, among which lineage B has been exclusively reported from other countries. Therefore, it is crucial to conduct uninterrupted timely molecular surveillance at the different geographical locations to learn about the evolutionary shift, DENV transmission, and key mutations in the DENV genome. This information will further help researchers to develop potential vaccine candidates against DENV. Furthermore, prior to the implementation of any vaccination trial, the surveillance of prevailing genotypes in the given area will serve as a key measure for the selection of appropriate prophylactics.

## ACKNOWLEDGEMENTS

We are thankful to Manoj Kumar and Ravi S. Singh for their valuable technical assistance in the field and laboratory. The authors express their gratitude to the government health officials, the Gorakhnath hospital technical staff, and all study participants. The authors would like to thank Mr Satish Ranawade for his technical assistance with nucleotide sequencing.

### Funding

This work was supported by the Indian Council of Medical Research, New Delhi (grant number GKP-1504) and the Department of Health Research, HRD, MoHFW, New Delhi, for Young Scientist fellowship scheme (DHR/YSS/000048). The funders had no role in study design, data collection and analysis, decision to publish, or preparation of the manuscript.

### Grant Disclosures

The following grant information was disclosed by the authors:
Indian Council of Medical Research, New Delhi: GKP-1504.
Department of Health Research, HRD, MoHFW, New Delhi: DHR/YSS/000048.

### Competing Interests

The authors declare that they have no competing interests.

### Author Contributions

- Sthita Pragnya Behera conceived and designed the experiments, performed the experiments, analyzed the data, authored or reviewed drafts of the article, original draft manuscript preparation, and approved the final draft.
- Pooja Bhardwaj performed the experiments, analyzed the data, prepared figures and/or tables, authored or reviewed drafts of the article, original draft manuscript preparation, and approved the final draft.
- Hirawati Deval conceived and designed the experiments, authored or reviewed drafts of the article, funding acquisition, and approved the final draft.

- Neha Srivastava performed the experiments, authored or reviewed drafts of the article, and approved the final draft.
- Rajeev Singh performed the experiments, authored or reviewed drafts of the article, data curation, Resources, and approved the final draft.
- Brij Ranjan Misra performed the experiments, authored or reviewed drafts of the article, data curation, Resources, and approved the final draft.
- Awdhesh Agrawal conceived and designed the experiments, performed the experiments, authored or reviewed drafts of the article, resources, and approved the final draft.
- Asif Kavathekar performed the experiments, authored or reviewed drafts of the article, and approved the final draft.
- Rajni Kant performed the experiments, authored or reviewed drafts of the article, review & editing, and approved the final draft.

## Ethics

The following information was supplied relating to ethical approvals (*i.e.*, approving body and any reference numbers):

The study was conducted following the guidelines established and approved by the Institutional Human Ethics Committee of ICMR-Regional Medical Research Centre, Gorakhpur for studies involving humans (RMRCGKP/EC/2022/3.17).

## DNA Deposition

The following information was supplied regarding the deposition of DNA sequences:

The data generated in this study are available at NCBI GenBank: MZ490477–MZ490537.

## Data Availability

The nucleotide sequences are available at NCBI: MZ490477–MZ490537.

## Supplemental Information

Supplemental information for this article can be found online at http://dx.doi.org/10.7717/peerj.14504#supplemental-information.

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
