# Peer review of "Co-circulation of all the four Dengue virus serotypes during 2018–2019: first report from Eastern Uttar Pradesh, India"

_PeerJ, doi:10.7717/peerj.14504_

## Round 0.1 · original submission · Major Revisions

This study of dengue viruses isolated from patients in 2018-2019 from Eastern Uttar Pradesh, India provides a useful snapshot of dengue types circulating at that time and demonstrates the spread of dengue types over time when compared with isolates sequenced from previously reported studies. But a number of concerns have been identified by two reviewers as well as the editor (acting as a reviewer) that need to be addressed before the manuscript can be accepted for publication.

Review-level comments from the Editor:
1. Please have the paper edited to improve English grammar.
2. The abbreviation for Uttar Pradesh within the abstract needs to be written out.
3. Line 17: Does “concurrent infections” refer to individuals co-infected by two different dengue virus types? If so, this is not referenced or discussed in the body of the paper other than on line 234 where it indicates that co-infection was not observed in the present study.
4. Grammar: The sentence on lines 41-43 needs to be rewritten to improve clarity.
5. Lines 50-52 repeats lines 38-40.
6. Please provide the PCR primer sequences in the Materials and Methods section.
7. Lines 108-110: Please provide the percentage of IgM and IgG positive samples for the 621 sequenced samples.
8. There are no figure legends!
9. The source of the clade designations for each dengue serotype is not provided. Please provide the references from which the clade designations have been derived.
10. For phylogenetic analysis visualization, in addition to the cladograms provided that show the clustering of the dengue isolates, you should also provide phylograms to show their genetic distance. These could be provided as supplementary figures (or the cladograms could be provided as supplementary figures, and the phylograms provided in the body of the manuscript.)
11. Parts of figures 1 and 2 has been cut off.
12. Figure 4 has been stretched and needs to be replaced with an unaltered figure.
13. It would also be useful to provide a single phylogram that includes all of the viruses reported in this study along with a few of the key viruses that serve as references for the various clades.
14. The newly sequenced isolates are identified in the paper and figures using internal reference numbers. It would be better to replace these numbers with the GenBank accession numbers of the individual sequences. Mapping of the reference numbers to accession numbers could then be provided as a supplemental table.
15. Lines 152-154. The relevance of this outbreak from 2021 to the data reported in this paper is not clear.
16. Lines 154-156. Make clear that the detection of DENV-2 in the E-UP was from past studies.
17. Starting at line 169, this paragraph refers to lineage I, II, and II of dengue virus. No lineage I, II, or III designations were provided in figure 2 or elsewhere in the paper for dengue 2 types.

Comments received from two additional reviewers are provided below. Please provide a response to all comments in a response to reviewers and please address the concerns in a revised manuscript.

Reviewer 1 ·

Basic reporting

1. This study, the authors identified the genotypes of dengue viruses spread in a E-UP area in Oct and Nov 2018 and 2019 by C-prM gene sequencing analysis of RT-PCR products from retrospective samples. The obtained information were compared with others previously reports in various neighboring areas to suggest dengue transmission and genotype emerging in E-UP.
2. The MS has some redundant sentences which should be modified. For examples in line 38-40 and 50-52; "During the past decades, the trends of DENV transmission has been shifted from urban to peri-urban and rural areas as well"
2. In abstract, the full term should be used in stead of abbreviation, as the reader may not understand. (for example; UP in line 19
3. The same words are sometimes inconsistency such as CprM or C-prM., core or capsid.
4. The overall written should be edited by native-English editors or professional language editing services.

Experimental design

1. In this study, the authors selected a prM-C gene region to see the heterogeneity of dengue serotypes/genotypes. Is it possible that the genotypes identified by this region may be different from those obtained by other gene regions such as E or NS1? If so, could you compare your results with others reported previously?
2. In general, NS1 antigen rapid test has lower sensitivity compared to RT-PCR. However, among 164 samples positive by rapid NS1 antigen test, only 61 are positive to RT-PCR. Could you explain what the possibility that make such a low positivity.

Validity of the findings

1. In this study, the authors could identify the apparent serotypes and genotypes of dengue virus in E-UP and see the correlation with the previous epidemics in other regions in India. It is fine to know, but they do not much generate specific or impact information to the fields as it has been accepted that dengue viruses are genetically heterogeneity.

2. The authors stated in the conclusion that "uninterrupted molecular surveillance around the year will serve as an early warning system to restrict the DENV outbreaks in this region. It is not convincing as molecular surveillance is costly and may not be fairly supported in mostly limited resources of dengue endemic areas. How rapid of the molecular results should be reported to public health authorities in order to prevent the outbreak in time? In my opinion, this information would be better beneficial to learn how far and rapid of dengue transmission or mutations from a place to place or from time to time.

Additional comments

The overall written should be edited by native-English editors or professional language editing services.

Reviewer 2 ·

Basic reporting

1. The English of the text is good.
2. Authors need to add more recent references regarding the RT-PCR method.
3. The article's structure is fine.
4. Results are good and relevant.

Experimental design

1. The research is original and primary, falling within the scope of the journal.
2. The research presents is well defined, presents relevant and meaningful data. The data contributes to scientific knowledge of the subject.
3. Uses robust but older methods.
4. The method used is well known and widely cited.

Validity of the findings

1. The authors present data with relative novelty in the area and with benefits for the literature.
2. Data were provided, but raw data related to sanger sequencing and RT-PCR of the same are not present. However, the data is usually requested by the NCBI so that the deposit of sequences can be made, once the authors made the deposited and the deposit data was delivered, I see no reason to request it.
3. The conclusion was very characteristic and linked to the topic in question.

Annotated reviews are not available for download in order to protect the identity of reviewers who chose to remain anonymous.

---

## Round 0.2 · Minor Revisions

Thank-you for your revised manuscript. We will be pleased to accept your manuscript for publication after a few minor details are addressed:

1) Please check, and update as necessary the apparent typo identified by reviewer 1 on line 221.

2) None of the GenBank sequences are available publicly. Please have these released and made public.

3) The figures provided are low resolution. Please submit higher resolution figures to PeerJ prior to publication.

Reviewer 1 ·

Basic reporting

No comment

Experimental design

No comment

Validity of the findings

No comment

Additional comments

The revised manuscript has been drastically improved in term of rationale and the impact of study to the field. The overall writing is more professional.

Remark: In the MS_rev1_clean.docx, line 221: "our findings showed that all DENV-3 sequences from this study" Does the author mean to DENV-1?

Annotated reviews are not available for download in order to protect the identity of reviewers who chose to remain anonymous.

---

## Round 0.3 · accepted · Accept

Thank-you for your quick response to our remaining concerns. I am happy to accept your manuscript for publication in PeerJ.